# Civil Service, HR Potential, and Open Innovation

**Aleksandra Polyakova**

Graduate School of Public Management, Russian Presidential Academy of National Economy and Public Administration (RANEPA), 119571 Moscow, Russia; polyakova-ag@ranepa.ru

**Abstract:** Countries strive to upgrade their civil service quality in many directions, thus responding to major contemporary challenges and following the trends of open innovation. The 2030 civil servant will require a wide set of soft and hard skills that are not common among the today's public administrators but are attributable to young people. The following questions then arise: Will the demand for qualified civil servants be reduced because of optimization of public administration processes? How significant is the demand for young people by the civil service? Is civil service attractive as a vector of career development among young people? The research objective is to provide quantity estimates of the Russian civil service's hiring potential and to match that potential with young people's attitude towards public administration employment. Using the panel regression, regions that can expand the civil service staff were identified and the necessary preconditions and policies to take use of this potential were indicated. The main conclusion is that governments should maintain incentives to join the civil service and improve the image of civil service among the youth to make it an attractive employer.

**Keywords:** civil service; public administration; young people; human resources potential; social network analysis; panel regression; labor market; open social innovation

## 1. Introduction

The human resource potential of the civil service is defined as an available and reserved manpower capable of solving current and future tasks of the civil service considering new conditions of functioning, challenges, and threats [1]. Reproduction of human resource potential means creation of conditions for continuous provision of the civil service with personnel of actual and advanced qualification through measures of professional development (internal reproduction) and import of professionals from other sectors (external reproduction).

The modern vector of the civil service's human resources potential development turns towards greater flexibility and adaptation of modern management practices that require civil servants to hold several specific competencies [2]. At the same time, the question arises: either to develop these competencies of existing employees, overcoming resistance to change, or to attract the holders of the necessary competencies—young people, whose talents will provide a synergetic effect of renewal of the civil service. The COVID-19 pandemic has put the civil service, as well as many other sectors of economy [3], in front of the new challenges that require reengineering many public administration processes, and which will result in increased efficiency of public administration. Existing civil servants will have to adapt to new realities—new project approaches, agile teams, a higher level of autonomy in decision-making, and a high level of automation and informatization, including in interaction with the beneficiaries of public policy—population and business. Without the involvement of young people, bureaucratic structures will rapidly lose effectiveness due to growing resistance to change, the propensity for which increases significantly as employees age. Overcoming this resistance will require significant financial and time costs.

Young people are strategically important for countries' economic development. Issues related to youth policy are quite deeply integrated into strategic planning documents, but the civil service is poorly integrated into the national system of identification and development of young talents and does not use the resources of the digital economy to identify and attract the most promising youth representatives. At the same time, in practical discourse, the feasibility of special measures aimed at attracting young people to the civil service is questioned and does not seem obvious considering the potential reduction in the number of civil servants as a result of increased efficiency, digitalization and optimization of administrative processes [4]. Verification comes from statistical data on the number of civil servants in Russia. It allows us to conclude that the civil service rejuvenation is not among the priorities of human resources policy: from 2013 to 2019, the number of civil servants under 30 years of age decreased by more than 30%, while the total number of civil servants decreased only by 3%. This situation undermines the demand for an influx of new employees, but on the other hand, the civil service will require people with a specific set of competencies that are more characteristic of young people. In this regard, the following issues become relevant:

- Will the demand for qualified civil servants be reduced because of optimization of public administration processes?
- How significant is the demand for young people by the civil service?
- Is civil service attractive as a vector of career development among young people?

Thus, the research objective is to provide quantity estimates of the Russian civil service's hiring potential and to match that potential with the young people's attitude towards public administration employment. The expected contribution of the objective to theory and practice includes additional quantifiable arguments to develop and implement incentive-based policies for human resource management in civil service. It also contributes to the practical implementation of the performance-based evaluation and assessment of public administration in terms of per-employee output of the civil service.

To achieve that, the following tasks are proposed:

- to identify the presence of civil service ageing in Russia,
- to assess the competitiveness of civil service as an employer,
- to find the efficient frontier between a region's population and the number of civil servants in the region,
- to analyze regional distribution of civil servants in terms of relative excess or deficit, thus, to identify the prospects of the recruitment demand in regions,
- to test the methodological approach to identification of the young people's attitude towards civil service employment.

The following research hypothesis is formulated: the hiring potential of the Russian civil service, as represented by the possible excess or deficit of civil service employees around the efficient frontier, is positive and can be taken use of, if public administration is relatively competitive as an employer.

## 2. Theoretical Background

To identify the young people, it is necessary to refer to national and international regulations. According to the UN approach, youth is the cohort of population aged 15–24 years [5]. The Russian regulation of youth policy traces back to 1994 when the federal program "Youth of Russia" was deployed. This document did not have a direct definition of the youth, but the youth were meant to be the people aged 15–29 years old. The current legislation initiatives in Russia tend to raise the upper boundary of youth to 35 years of age.

Issues of youth employment have been within the scope of research due to the many aspects of the problem. E.g., Lewis and Hayes [6] look through the changing pattern of youth employment in Europe and come to find there is an increasing shift from permanent full-time to temporary part-time

contracts accompanied by the 'hollowing out' of traditional mid-skill level occupations. They raise questions concerning the most appropriate policy approaches to education and training to enable young people's career development. Another issue is being addressed by A. Holford [7] who studied labor market outcomes for the young people who took advantage of part-time work during their compulsory education.

O'Reilly, Grotti and Russell [8] made a cross-sectoral study of industries and trades' demand for youth workers. They found that "the decline in youth employment observed with the Great Recession does not seem to be driven by structural forces such as the shrinkage of some sectors, but rather by an overall lower likelihood to employ young people in particular sectors", and proved that public administration was much less youth-intensive compared to HoReCa, retail, etc. Earlier works, e.g., Pilichowski, Arnould, and Turkisch [9] also prove that the Organization for Economic Cooperation and Development (OECD) labor market is characterized by a "far smaller proportion of young employees in the public sector than in the private sector". Remeikiene et al. [10] found that youth overall unemployment in the European Union (EU) countries is facilitated by such factors as the reluctance of businesses to employ unqualified or low-skilled young people.

Critical investigation of the civil service's role in employing the young people in distressed economies is provided by T. Acheampong [11], who described a phenomenon of "the Government as Employer of Last Resort". Such an approach is determinant in the policies of several European countries, where the youth employment by public administration is found to be driven by the following objectives:

- to improve the human resource potential of the civil service and to increase the quality of public administration by hiring young people,
- to combat poverty, social inequality and unemployment by training and hiring young people to the civil service,
- to combat civil service ageing.

The listed objectives provide a generalized answer to the question about reasons for hiring young people to the civil service.

A number of contemporary studies convincingly prove that young people are much more self-competitive, more adaptive, and more receptive to new technologies, including managerial ones, and therefore, without involving young people, bureaucratic structures will rapidly lose effectiveness due to growing resistance to change, the propensity for which increases significantly as workers age. Therefore, the implementation of the project approach in the civil service, progressive adaptive decision-making, and product development methodologies [12], such as Agile, will be much more effective due to the lack of established practices among young people; change is usually resisted by employees of bureaucratic structures. There is also considerable potential for savings in attracting young people in the retraining and professional development of civil servants, especially in the development of soft skills. Along with the emerging new challenges to the public administration system, youth involvement is becoming a critical task.

There are a few studies dealing with matters of public administration attractiveness as an employer, especially for the young people. Several recent studies raise issues like gender inequalities that might determine the unattractiveness of civil service career. Isupova and Utkina [13] show that young Russian women are not satisfied with routine work and low salaries and, as a result, try to find positions with higher salaries or better conditions, that are frequently not associated with civil service. Utkina and Gasparyan [14] made an experiment that reveals that the civil servants responsible for interactions with job applicants tend to reproduce gender stereotypes: young women receive a motherhood penalty upon entry into civil service due to dominating masculinity. A piece of contrast is brought in by G. Lewis [15], who found that the American graduates with bachelor's degrees in public administration/policy are among the most likely to choose public sector employment and earn more than comparable graduates in most competing fields. He concludes this could become a factor to support employment attractiveness

of civil service, even though the young often consider public administration as an employment of last resort.

Social studies and public surveys can provide necessary information on the young people's attitude towards public administration, but standard approaches are costly, time consuming, and their results can be significantly biased. More importantly, standard approaches do not allow for identification of proper channels to influence the opinion of youth. The latter issue means it is necessary to discuss the possible ways to change their attitude towards public administration and to attract the most skilled ones. To do that, the civil service needs to improve its general image and to employ technologies to find and attract the best candidates. The traditional marketing in the labor market has limited potential to influence the younger minds. Social network analysis can be employed to identify major influencers and to direct the proper information signals to the necessary audience.

Recently, social media has become an increasingly common source of information acquisition and dissemination. They have found their application in a variety of fields and most often their experience is disclosed in the literature. The growing involvement of the population in social networks opens up new opportunities for analyzing different communication models. For example, data from social networks can be used to identify problems in the social environment, trends, to find influential actors and other types of information. For example, C. Golder and M. Macy [16] monitored the behavior of individuals on Twitter and studied how people's moods change over time of day, day of week, and season.

A separate layer of studies is devoted to solving social problems by analyzing related events based on big data. They reflect research methods, techniques, and algorithms that enable researchers to implement them to solve problems affecting both individuals and society. Solutions to such problems can be achieved by measuring public opinion and identifying signs of harmful behavior through predictive analysis. In particular, the study by Kursuncu et al. [17] shows the ways to identify and predict different crimes and malpractices such as cyberbullying, etc. Criminal communities use social networks to pressure their competitors, and identifying such users helps law enforcement agencies anticipate a crime before it can happen. For example, the work of Balasuria et al. [18] reflects the problem of finding street gang members on Twitter. In their research, almost 400 gang member profiles were manually identified using initial terms, including gang related rap-performers, their retweets and subscribers using the tweet text function, YouTube video descriptions and comments, smileys and image profiles to work with various machine learning algorithms, including Bayesian algorithm, logistic regression, and so on.

Thus, the digital economy has the potential to improve the competitiveness of the civil service in the labor market. Resources of digital economy can be used to assess the image of civil service as a segment of the labor market, to identify positive and negative trends in its perception among young people, to assess the mismatch between expectations and reality, to increase its attractiveness, as well as to identify the most promising candidates to attract young people to civil service.

The key idea is to extract information from the digital space that can directly or indirectly point to subjective opinions of individuals affecting the civil service as a potential vector of a young person's career development. This is based on social network data and processing it with tools of social network analysis. The latter allows systematizing unstructured data, identifying key actors that form public opinion, as well as channels for its dissemination.

This task of key actor identification has a number of solutions obtained in economics, sociology, linguistics, and other fields of knowledge. However, the most rational method was suggested by Kolya et al. [19] who developed and described a way to formalize the links between actors, taking into account the characteristics of these links. Of the earlier works, the study by M. Farrugia and A. Quigley [20], who investigated a number of ways to build networks of interaction between actors and described approaches to identifying opinion leaders, deserves attention.

## 3. Materials and Methods

To address the research hypothesis, the following research design is proposed.

First, the current civil service staff is analyzed regarding its age structure (Section 4.1). Our analysis employed the statistical data published by the Russian Federal Statistics Service (Rosstat).

The second stage deals with the civil service's attractiveness as an employer (Section 4.2). The research will address the comparison of nominal average wages and their growth rates across economic sectors including civil service, since wage is also a factor in making decisions about building the career path of young people. Conclusions made on the growth rates and rankings, as well as absolute figure comparisons, can characterize the civil service distinctly and clearly as an attractive or unattractive career development path.

The third stage is devoted to estimating the hiring potential of the civil service in the Russian regions. Regression analysis is used to find an efficient frontier—the estimated excess or lack of civil servants in a particular region according to the distribution of population and civil service staff.

The main idea of the analysis is the following: if, according to the model, a region has an excess quantity of civil servants, then the civil service's hiring potential can be considered relatively low and the young people's chances of being recruited by the sector of public administration are much lower; alternatively, the estimated lack of civil servants can indicate the regions where the hiring potential is in favor of young people. The regression analysis is based on the following theoretical proposition: there is a specific quantity of civil servants which is enough to enable the functions of public administration and depends on a region's population. The efficient frontier can be obtained by regressing the number of civil servants with the population across the panel. Model values' fluctuations below and above the efficient frontier will reflect the lack or excess of civil servants.

The background for statistical modeling is the general trend to decrease the number of civil servants due to the many parallel processes taking place contemporarily:

- limitations and the new modes of labor organization due to the CoVID-19 pandemic,
- the long-standing request for optimization of public administration to increase the per-head productivity.

The two factors decrease the future demand for civil servants, including the younger ones. Still, given the Russian regions' differentiation [21], there might be territories where the potential to increase the number of civil servants is not depleted. Then, it is necessary to analyze data on the number of civil servants in regional executive bodies per 10 thousand people of the permanent population. This indicator is published by the official statistics and is considered one of the qualitative characteristics of public administration system efficiency.

The data on 83 Russian regions (except for Crimea and Sevastopol that do not have the prolonged time series) were processed to identify outliers. The four regions with the largest and the smallest population were excluded. The final data set included 6 periods (2013–2018) times 79 regions to comprise 474 cases.

The number of employees of regional executive authorities was obtained by multiplying the permanent population of a region by the number of civil servants per 10,000 people. The obtained figure was regressed against the same year population.

The linear regression model was used due to the supposed linear nature of the process. The model quality characteristics that were analyzed include R-squared, t-statistics, F-criterion.

Model values were compared with the actual ones; the relative differences and their distribution were used to characterize the outcome.

The fourth stage is devoted to assessment of the mismatch between the image of civil service as an employer and a young person as a potential employee. It is based on a research algorithm, similar to the one described in Lynn et al. [22] or in Kolmakov, Rusneva, & Thalassinos [23]: it employs content analysis of social media discourse, specifically—Twitter publications.

The final stage of the study is to combine the research outcomes and conclude on the necessity to attract young people to the civil service and to discuss further policy implications and proper theoretical grounds for it.

## 4. Results

### 4.1. Ageing of the Civil Service in Russia

From the point of view of the demand for civil service by young people, the following trend is noteworthy: along with a decrease in the total number of civil servants from 2013 to 2019 by 2.8% (federal civil servants) and by 3.1% (regional civil servants), the number of employees under 30 years of age suffered the greatest reduction—by 31.2% and 33.0%, respectively (see Table 1).

**Table 1.** Number of civil servants by age groups and levels of government.

| Federal Civil Servants | <30 y.o. | 30–39 y.o. | 40–49 y.o. | 50–59 y.o. | >60 y.o. | Mean | Total |
|---|---|---|---|---|---|---|---|
| as of 1 October 2013 | 155,246 | 180,567 | 116,863 | 98,529 | 13,285 | 38 | 564,490 |
| as of 1 October 2016 | 127,939 | 186,385 | 120,101 | 91,931 | 15,096 | 39 | 541,452 |
| as of 1 October 2019 | 106,840 | 192,179 | 137,878 | 88,952 | 22,866 | 40 | 548,715 |
| Growth rate 2019 by 2013, % | −31.2 | 6.4 | 18.0 | −9.7 | 72.1 | 5.3 | −2.8 |
| **Regional Civil Servants** | **<30 y.o.** | **30–39 y.o.** | **40–49 y.o.** | **50–59 y.o.** | **>60 y.o.** | **Mean** | **Total** |
| as of 1 October 2013 | 42,285 | 72,403 | 52,748 | 48,778 | 5696 | 41 | 221,910 |
| as of 1 October 2016 | 35,911 | 76,171 | 57,569 | 42,017 | 5655 | 41 | 217,323 |
| as of 1 October 2019 | 28,333 | 75,070 | 65,245 | 38,575 | 7832 | 41 | 215,055 |
| Growth rate 2019 by 2013, % | −33.0 | 3.7 | 23.7 | −20.9 | 37.5 | 0.0 | −3.1 |

Source: author's calculations.

Ageing of civil service is an obvious trend as the number of civil servants aged above 60 has increased at the federal level by 72.1%, and at the regional level—by 37.5% in five years. From 2013 to 2019, the largest age cohort of civil servants—30 to 39 years old—increased by 6.4% and 3.7% for federal and regional civil servants respectively, which is largely due to the age transition and indicates a decrease in the inflow of young people to public administration. A recent increase in the retirement age in the Russian Federation will reduce the staff outflow, which will also act as a factor limiting the demand for young personnel.

### 4.2. Civil Service as an Employer

Public administration sector competitiveness is assessed based on the study of the nominal accrued wages growth dynamics across economic sectors including "public administration and provision of military security, social security" (hereinafter—"public administration" or "state management").

In 2013–2018, the average annual growth rate of wages in public administration was 3.4%, while the average for the economy was 8.0% (Table 2).

Obviously, from the point of view of the average annual growth of wages, public administration is not the most attractive place to work. The growth rate in 2019 slightly increased to 6.75 but is still the lowest in the sample of economic sectors.

From 2013 by 2018, the average wage in the public administration sector increased by 18.2%, which indicates a rather moderate and regulated growth of civil servants' income, unlike some other sectors, where the growth of wages is regulated only by the demand and supply of qualified personnel.

**Table 2.** Growth rate of nominal accrued salary in some economic activities, including public administration.

| Economic Trades, Selectively | 2013–2018, Annual Average | 2018 to 2013 | 2019 to 2018 |
|---|---|---|---|
| Total in the economy | 8.0 | 46.8 | 9.5 |
| Communication and IT | 16.3 | 112.4 | 14.0 |
| Agriculture, hunting and forestry | 12.8 | 82.5 | 10.6 |
| Health care and provision of social services | 10.4 | 63.8 | 7.7 |
| Mining of minerals | 9.0 | 53.6 | 7.4 |
| Wholesale and retail trade; repair of motor vehicles, motorcycles, household goods and personal items | 8.9 | 53.0 | 13.2 |
| Processing industries | 8.5 | 50.6 | 7.7 |
| Education | 7.9 | 46.5 | 7.9 |
| Hotels and restaurants | 7.5 | 43.4 | 7.4 |
| Financial services | 7.5 | 43.8 | 13.8 |
| Construction | 6.8 | 39.0 | 10.7 |
| Public administration and provision of military security; social insurance | 3.4 | 18.2 | 6.7 |
| Operations with real estate, rent and service provision | −0.4 | −2.2 | 11.4 |

Source: author's calculations.

Partly, low growth rates of average accrued wages in public administration can be explained by the effect of a high basis: in 2013–2015, public administration was ranked third in terms of average accrued wages, right after financial services (1st place) and mining (2nd place), which largely determined the popularity of the civil service among job seekers. Since 2016 the public administration has been in fourth place, after "Communication and IT".

As of 2019, the average salary in public administration was 1.8 times higher than the lowest in the sample sector's, while in 2013 the gap was 2.6 times. Compared to the average in the economy, the gap has narrowed from 136% in 2013 to 107% in 2019, which indicates the alignment of the average accrued salary across economic sectors. This indicates a relative decline in the civil service competitiveness in terms of income and the dynamics of its expected growth. The latter argument is supported by the continuing growth of the gap between average wages in public administration and in the maximum in the sample sector's wage: in 2013, the average accrued wages in public administration were 64% of the maximum, while by 2019 they decreased to 49% of the maximum in the sample (see Figure 1).

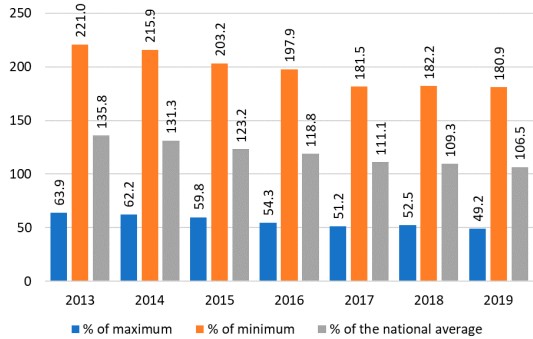

**Figure 1.** Relative characteristics of average salary in public administration. Source: author's calculations.

The described gap is expected to increase, because the dynamics of wage growth in public administration, as shown above, is one of the lowest. Correspondingly, the rank of the civil service in terms of the average wage will also decrease. Hence, there is another significant problem: as educational credits spread, the starting salary and its subsequent growth rates will be the major factors determining a career choice. The latter means that civil service will become less attractive for individuals without work experience.

At the same time, it is necessary to note a significant fact, which has been verified by several crises, including the large-scale and global limitation of economic activity provoked by the COVID-19 pandemic. Practice has shown that the public service, like a number of budget-funded activities, has proven to be a "safe haven" in the labor market, as the government, as an employer, is better placed to secure its obligations to employees. This fact is also a factor in the competitiveness of the civil service in the labor market—a kind of conservative strategy of personal development in terms of financial security and stability.

Thus, the state civil service as an employer of young people is gradually losing its attractiveness. From the point of view of the civil service, the interest in attracting young people is also not obvious.

### 4.3. Statistical Analysis of the Growth Potential of the Civil Service Human Resources across the Russian Regions

As of 2018, the Russian regional executive authorities had an average of 11.31 civil servants per ten thousand people of the permanent population, and the value of this indicator has decreased by 5.8% since 2013. At the same time in 26 regions the number of regional executive authorities has increased. The highest relative growth was recorded in Stavropol Krai (16.0%), Altai Krai (15.9%), Perm Krai (15.6%). Nevertheless, the number of civil servants in these regions does not exceed the Russian average (Table 3).

**Table 3.** Dynamics of per capita number of civil servants of the Russian regional executive authorities (select regions).

| Region | Number of Regional Civil Servants per 10,000 of Population | | | Permanent Population in 2018, People | Regional Civil Servants Estimate in 2018 |
|---|---|---|---|---|---|
| | **2013** | **2018** | **Growth Rate %** | | |
| Stavropol Krai | 6.7 | 7.77 | 16.0 | 2,797,958 | 2174 |
| Altai Krai | 9.7 | 11.24 | 15.9 | 2,341,447 | 2632 |
| Perm Krai | 8.4 | 9.71 | 15.6 | 2,616,961 | 2541 |
| Republic of Dagestan | 10 | 11.38 | 13.8 | 3,075,006 | 3499 |
| Kirov region | 11.1 | 12.43 | 12.0 | 1,277,673 | 1588 |
| Khanty-Mansiysk Autonomous Okrug | 15.2 | 16.82 | 10.7 | 1,659,435 | 2791 |
| Republic of Khakassia | 13.9 | 15.31 | 10.1 | 536,840 | 822 |
| … | | | | | |
| Russian Federation national average | 12 | 11.31 | −5.8 | 146,830,576 | 166,065 |
| … | | | | | |
| Tyumen region | 17.7 | 14.2 | −19.8 | 1,508,737 | 2142 |
| Ulyanovsk region | 11.2 | 8.47 | −24.4 | 1,242,517 | 1052 |
| Kostroma region | 21.8 | 16.48 | −24.4 | 640,296 | 1055 |
| Republic of Bashkortostan | 12.2 | 9.14 | −25.1 | 4,057,149 | 3708 |
| Mari El Republic | 17.6 | 13.17 | −25.2 | 681,357 | 897 |
| Chechen Republic | 26.5 | 19.75 | −25.5 | 1,446,966 | 2858 |
| Udmurt Republic | 12.3 | 9.11 | −25.9 | 1,510,217 | 1376 |
| Kurgan region | 17.7 | 12.32 | −30.4 | 840,119 | 1035 |
| Moscow region | 10.5 | 6.46 | −38.5 | 7,551,516 | 4878 |

Source: author's calculations.

More remarkable are the dynamics of reduction of this indicator. The leader of the reduction in 2013–2018 is the Moscow region (by 38.5%), another 26 regions showed a reduction of more than 10%. As of 2018, the Moscow region was among the three regions with the lowest number of civil servants in the regional executive authorities (6.46 per 10,000 people), approximately the same in Rostov region (6.66), and the least in Chelyabinsk and Tula regions—5.72 and 5.07 respectively.

Regression of the civil servants by a region's population across the years and in the bulk panel shows that the models are statistically reliable and stable in time (see Table 4).

**Table 4.** Regression results.

| Coefficient | 2013 | 2014 | 2015 | 2016 | 2017 | 2018 | The Panel |
|---|---|---|---|---|---|---|---|
| intercept value | 432.01 | 429.42 | 411.69 | 399.78 | 370.15 | 354.55 | 399.73 |
| t of intercept | 3.80 | 4.00 | 3.27 | 3.86 | 3.69 | 3.73 | 9.14 |
| p of intercept | 0.000 | 0.000 | 0.002 | 0.000 | 0.000 | 0.000 | 0.000 |
| b (×10,000) | 8.8769 | 8.7854 | 8.7897 | 8.6296 | 8.7672 | 8.7127 | 8.7594 |
| t of b | 15.3 | 16.1 | 13.7 | 16.4 | 17.2 | 18.1 | 39.4 |
| p of b | 0.0 | 0.0 | 0.0 | 0.0 | 0.0 | 0.0 | 0.0 |
| Model R squared, % | 75.21 | 77 | 70.93 | 77.74 | 79.4 | 80.92 | 76.65 |

Source: author's calculations.

The regression coefficient of the argument can be interpreted as an eight to nine civil servants quantity increase in response to every increase of 10,000 in the population. The model describes at least 71% of variance of the values.

In order to draw conclusions about the future dynamics of increasing the number of civil servants in the Russian regions, it is necessary to study the distribution of specific characteristics in the coordinates of "population—estimated number of civil servants in the regional executive authorities".

The scatterplot (Figure 2) allows us to estimate a trend line—the efficient frontier of the number of civil servants in the regional executive authorities with respect to population. Several conclusions can be drawn: if a region is below the trend line, one can assume that the potential for growth in the number of civil servants has not yet been depleted, and if it is above the trend line, the number of civil servants in regional executive authorities is relatively excessive.

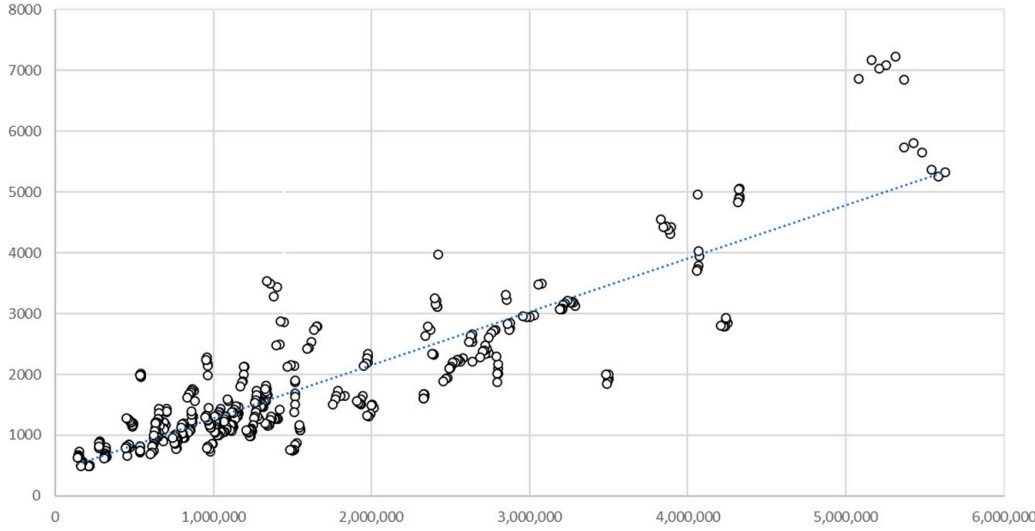

**Figure 2.** Distribution of Russian regions by population and estimated number of civil servants in regional executive authorities. Source: author's calculations.

The figure illustrates the relative excess of the number of civil servants of the regional executive authorities over the efficient frontier in Khanty-Mansiysk and Yamalo-Nenets Autonomous Districts, St. Petersburg, Irkutsk, Tyumen, and Vologda regions. A number of other regions are formally above the trend line, but within the confidence interval.

The regions' distribution around the efficient frontier is not homogenous: 44 of 79 regions are below the trend line which means they have potential to hire people to regional executive authorities (see Table 5).

**Table 5.** Distribution of regions.

| Parameter | 2013 | 2014 | 2015 | 2016 | 2017 | 2018 |
|---|---|---|---|---|---|---|
| Quantity of regions below trend line | 44 | 46 | 50 | 44 | 46 | 44 |
| Quantity of regions above trend line | 35 | 33 | 29 | 35 | 33 | 35 |
| Maximum excess, actual value in % to model value | 218.8 | 217.4 | 318.6 | 229.8 | 237.6 | 243.8 |
| Maximum potential, actual value in % to model value | 48.7 | 48.2 | 43.4 | 44.3 | 45.4 | 45.7 |
| Median region, actual value in % to model value | 95.1 | 96.8 | 96.4 | 97.5 | 97.3 | 98.4 |
| Average actual value in % to model value | 97.9 | 98.4 | 97.1 | 98.3 | 98.7 | 98.8 |
| Standard deviation, % actual value in % to model value | 31.6 | 30.8 | 36.1 | 30.9 | 30.1 | 29.6 |
| Quantity of regions with actual value in % to model value within [95%; 105%] | 10 | 15 | 18 | 21 | 22 | 19 |

Source: author's calculations.

In 2018 several regions had the potential to double the number of civil servants, even though the record excess amount is also significant. The median region needed civil servants; the average to median value is very harmonized, even though the model values are significantly volatile. By 2018 the number of regions are close to being balanced; civil servants' quantity to population ratio reached 19.

Namely, Rostov and Chelyabinsk Oblasts, Krasnodar Krai, and the Republic of Bashkortostan have the potential to increase the number of civil servants, provided the current paradigm of civil service development is maintained. The regions furthest from the trend line from below may also have a higher efficiency of public administration in terms of managerial labor productivity.

Thus, the civil service in regional executive bodies located below the trend line is supposedly more open to young people because of its potential for growth or to maintain higher efficiency.

*4.4. Image Evaluation of Civil Service*

To test the potential of social network analysis, a set of 30 words and phrases was formed. Semantic core was obtained by the content analysis of search query frequency of the key phrases and their forms: "civil service", "public administration", "civil service employment", "civil servants".

All Twitter messages from June 01 to September 01, 2020 containing the key phrases from the core were retrieved using NodeXL software. The Russian-speaking segment of the Internet was studied, and posts were censored by geographic origin (posts by authors not from Russia were excluded). The censored data set contained 1447 posts and reactions to them. About 20 clusters were determined in the data; still, the topics of discussion within clusters were not unique. Thematic analysis of clusters highlighted the most significant areas of interest of actors in relation to certain aspects of civil service:

(1) legislation on the state civil service—28 original messages,
(2) civil service personnel reserve—45 original messages,
(3) public administration employment—36 original messages. These messages and the reactions of users to them—the further dissemination of the message across its subnetwork—form the most significant informational impulse in the data sample.
(4) mention of civil service in other contexts—144 original messages.

The sample analysis shows no evidence of significant "centers of influence". Still, the discourse analysis is quite informative in terms of specific attitudes of the people towards public administration employment. All the posts are in Russian language.

The greatest response had the post of @AShamilov containing the link to a business journal article: "Taliya Minullina: If an expensive brand's bag is your objective, then the civil service is not for you. Officials should not complain about low salaries; they just need to manage funds wisely and have higher goals and values". Such a citation had mostly negative comments.

The next popular post was by @strmhnd, who wrote that she intended to quit civil service and regretted that the new job offer was also in public administration: "Comrades! Unbeknownst to my current bosses, I am going for an interview today. Interesting position, but still a civil service. Wish me good luck!". The motif of regret with relation to civil service is very indicative in determining the general attitude of the Russian young people towards civil service.

Several more characteristic phrases are worth being mentioned:

"Work in public administration is [disaster], because in fact, for a normal life you need to create a workplace for yourself, and everything that is offered in vacancies is just a mockery for a person, whose level of ambitions is somehow above zero"(@nofoundgirl)

"Lawyers seem to have a slightly better situation, but on the whole, they are about the same. Especially if lawyers employed by civil service, then the same darkness"(@chubby_bunny_25)

"Since ancient times, it has become a custom, whoever studied the worst, has only one perspective—the civil service"(@Negotov)

Eventually, there is no evidence of a systematic policy aimed at forming the image of civil service among young people. According to the dominant network discourse, work for the civil service is mentioned only in the context of the problems of its legal regulation and difficulties associated with working in government bodies.

## 5. Discussion: Civil Service and Open Innovation

Obvious evidence of the Russian civil service ageing was obtained. This is not a problem by itself, since the professionalism in administration is often the function of experience, which means the older the personnel of civil service, the higher the quality of public administration. Of course, such a conclusion is subject to verification.

From the age structure point of view, the Russian civil service does not seem to have an obvious priority to rejuvenate public administration, as several European countries do: Sweden, which is experiencing a period of significant population ageing, attracts young people to the civil service in order to solve the problem of staff aging in certain government bodies that need to change their age structure [24].

Still, public administration is getting closer to facing challenges of open innovation, as well as its benefits. The paradigm of open innovation presumes that collective actors, such as civil service bodies and public organizations, will become more sensitive to external innovations in terms of adaptation and implementation. Another tier of open innovation influence is represented by the so-called creative consumers and their communities driven by innovations. The younger cohorts of the population are traditionally more innovation prone. Their attitude towards civil service can influence the progress of public administration significantly in both directions. Social capital and the youth's intellectual potential can contribute sufficiently to the triple helix of innovation to make it a quad-helix. In a generalized context public administration is not a competitive and attractive employer [25], contrary to business and R&D centers. What is more important, civil service is associated with many restrictions that undermine its competitiveness from the young people's point of view. In many OECD countries the public sector is not leading the labor market in terms of wages, but it offers significant non-cash benefits; the same is true in Russia. This remains the major factor of the civil service's competitive

advantage. The ongoing discussion to decrease incentives to civil servants has rational grounds, but from a strategy point of view the benefits should be maintained. The national and regional youth policies must consider the possibility of "employer of last resort" phenomenon, which is in place in many territories worldwide. This must not become the discriminating factor, otherwise it will foster the migration outflows from the distressed territories.

Given the trend of open innovations driven technology implementation in every sector, the future demand for employees is going to be suppressed. The public administration will probably be the last to face this threat, but this will inevitably influence the hiring trends and career patterns in civil service. Forward-looking steps have to be taken today to redesign the educational programs in favor of the universality and cross-sectoral applicability of skills of public administration degree holders.

The efficient frontier derived from regression analysis can be used to assess the relative per-employee performance of the civil service if related to any economic, demographic, or other indicator. The extremum values of civil servants' excess, attributable to several Russian regions, can be explained economically. Most of the "excess" regions are the distressed ones, they have significant levels of unemployment: up to 25% compared to the national average of 5–6% in the beginning of 2020. Eventually, the civil service acts as the employer of last resort in many regions of Russia. If the analysis is reproduced on other countries' data or on the cross-national panel, its results might provide another indicator for measuring economic and labor market distress.

The cases of regions that "lack" civil servants require deeper investigation to resolve the dilemma: whether public administration in those regions is highly productive in terms of per-employee performance, or the efficiency of public administration is undermined by the lack of professionals and can be improved by importing the most mobile cohort—the young people—by providing additional incentives and stable career prospects.

Again, such a study methodology, if it used on another country's data, can be integrated in regional and national policies of labor market regulation and education programs to increase the strategic resilience of civil service and to assist in avoiding crises, at least in the public sector of employment.

Taking into account the specific features of the current situation, one can expect a downward shift of the efficient frontier in response to optimization processes caused by the coronavirus pandemic and the impulse of transition to remote and more effective work formats due to the restrictive measures taken. In this case the majority of regions will be above the trend line, i.e., with an excessive number of civil servants in their executive bodies. The conclusions from such a forecast are ambiguous: on the one hand, such a development undermines the demand for an influx of new employees, but on the other hand, the civil service will require people with a specific set of competencies, hard skills, and soft skills that are more characteristic of young people.

In line with the mentioned "youth-and-civil service" policies, the Russian Federation policy to deal with the two issues needs to address the two possible objectives: to use civil service to support employment, and to attract the young men and women to enable the public administration quality increase.

A brief, "introductory" use of social network analysis—another product of open innovation—provides at least two outcomes. Firstly, it is becoming obvious that a profound and longitudinal study of social network discourse is necessary to obtain deeper insights into the problem of young people's attitude towards civil service employment, which is presumably negative. Secondly, the governments of all levels can be recommended to implement strategies aimed at maintaining a positive or, at least, neutral image of the civil service in the network discourse, since social media is becoming the most influential platform in making and propagating opinions.

## 6. Conclusions

### 6.1. Implication

The civil service in Russia is ageing, along with the civil service all over the world. More importantly, it requires new skills that are difficult and expensive to develop and much easier to obtain from the outsource supplier—the young people who intrinsically possess the necessary skills or, at least, can be educated with the less effort, as Reza et al. [26] see it. The civil service is under-competitive compared to several other economic sectors which means it will require additional resources to attract, hire, and retain young people. The optimal way to achieve that goal is to use the potential of big data and social network analysis to identify the key actors that influence individuals' opinion with regard to any common problem, including civil service employment.

Another important issue that can be derived from the regression analysis made is the potential direction for cross-regional human resources mobility in response to productivity growth challenges and new managerial practice implementation.

Obviously, the human resources potential of the civil service can be developed by attracting young people. The sources of synergy lie in the search for talents and attracting them to the civil service. For this purpose, a methodological approach to identifying the most promising candidates for the civil service from among young people is proposed. This approach is based on big data exploration and social network analysis, which helps to identify implicit dependencies, like in Herningish et al. [27], and opinion leaders, identification of which will allow for more effective communication between the civil service and young people in planning human resources policy.

### 6.2. Limits and Future Research

The research hypothesis was confirmed: the hiring potential of the Russian civil service is rather moderate, even though many regions face a deficit of personnel as measured by the efficient frontier. An average Russian region has the potential to increase the number of civil servants by 1.5% which means that chances for young people to get employed by public administration depend on two factors: competing opportunities in the labor market and the personal attitude towards public sector employment. The first issue is purely economic and is generally quantified in this paper, while the other issue has to be addressed more precisely using data analysis algorithms that employ social network analysis and other techniques. The latter is the most evident research prospect resulting from this paper.

The paper contributes to the theory with a methodology of the efficient frontier definition, estimation, and interpretation rule, that can facilitate further developments of theoretical grounds for performance assessment in the public sector across cities, regions, or countries. The paper findings can be instrumental in policy design, including human resource, youth, education, and economic policy.

**Funding:** The article was prepared as part of the research "Reproduction of human resources potential and attracting young people to the civil service" within the 2020 state assignment to the Russian Presidential Academy of National Economy and Public Administration.

**Acknowledgments:** The author would like to thank the anonymous reviewers whose expertise, valuable comments, and suggestions contributed to the manuscript improvement.

**Conflicts of Interest:** The author declares no conflict of interest. The funders had no role in the design of the study; in the collection, analyses, or interpretation of data; in the writing of the manuscript, nor in the decision to publish the results.

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
