# Peer review of "Civil Service, HR Potential, and Open Innovation"

_2199-8531, doi:10.3390/joitmc6040174_

Round 1

Reviewer 1 Report

Having read the current version of the article, I believe that the amendments made increased the scientific value of the text. A clearly stated purpose in the introduction, acceptable hypotheses, and a broader review of the literature lead to the conclusion that the article can be published.

Author Response

Dear Sir / Madam,

I am greatful to have your appreciation of the revised paper.

Your valuable remarks and suggestions encourage me to continue our research.

Thank you!

Reviewer 2 Report

The proposals of various reviewers were well revised and supplemented.

You've had a hard time.

Author Response

(The authors gave the same response as above.)

Reviewer 3 Report

Thank author for improvements. All comments were incorporated and paper was significantly improved.

Author Response

(The authors gave the same response as above.)

Reviewer 4 Report

Dear Authors!

Thank you for taking up a very interesting topic of civil service HR-potential and the youth. It is certainly an issue that deserves investigation.

I appreciate the work done in improving the manuscript, its quality is much better than the original version, but the text is still not free from a few flaws.

The list of main shortcomings to be removed is presented below:

  • The research questions posed in the abstract should be identical to the questions presented in the introduction /lines 54-57/. They are not the same in my opinion.
  • The conclusions should contain clear information as to whether the research hypothesis /lines 74-76/ has been confirmed or not.
  • There is an incorrect numbering of tables. Table 3 is missing.
  • In the "materials and methods" section in the description of the fourth stage of research /lines 228-230 / there should be a sentence saying that content analysis of Twitter was carried out. Then the description of the research methodology used will be more complete.

After removing a few indicated defects, I can recommend the text for publication.

Regards,

The reviewer.

Author Response

Dear Sir / Madam,

I am greatful to have your appreciation of the revised paper.

Please, be sure I made the following amendments:

1) I synchronized questions posed in the abstract with the ones in Introduction. Now they are similar which is undoubtedly more clear.

2) The Conclusion text was slightly corrected to link the appropriate pragraphs with the research hypothesis. The third parapgraph of the Conclusion now indicates the hypothesis found its proof.

3) The tables were renumerated. Thank you. 

4) As for "Matrials and methods", I agree it is better to indicate the social medium used. Proper text is now in place.

Your valuable remarks and suggestions encourage me to continue our research.

Thank you!

This manuscript is a resubmission of an earlier submission. The following is a list of the peer review reports and author responses from that submission.

Round 1

Reviewer 1 Report

The problem discussed in the article is important and undoubtedly important. However, significant weaknesses should be noted.

  1. Firstly, the introduction lacks a clearly defined purpose of the article, which makes it difficult to assess its results.
  2. Second, the article does not formulate hypotheses, which causes the statistical analysis to lose value.
  3. Thirdly, the article is local in nature, from which it is difficult to draw general conclusions.

Reviewer 2 Report

Thank you for giving me the opportunity to review "Civil service and the young people: issues of synergy creation in human resources potential." This study was well written. There is only one thing to supplement.

Title: the title does not adequately reflect the content of the paper. Please, try to change it to better inform the readers about that origin of your sample. The impact of local conditions (Country, culture, economy) could influence in a different way.

Reviewer 3 Report

The theme of the paper is interesting. However, the paper does not meet the quality standards and needs significant improvements.

The whole paper is descriptive in nature and does not bring any new findings other than reflection on current situation. The most important points to be improved are the following:

  • Theoretical background is missing totally. Currently, just a short introduction is given. But deeper theoretical grounds for the analysis is not provided.
  • The theoretical background does not specify hypotheses and how theoretical background leads to elaboration of this specific research, methods, and approach.
  • The aim is not clear and grounded in current theory and approaches. It does not refer to the journal scope. It is not clear what the analysis in this paper should bring to the reader. It is just analysis without implication. Please, add implications and benefits for readers and work more in context in definition of your aim.
  • Methods does not contain important information such as sample of survey used, its demographic characteristics, how were data from interviews analysed and interpreted, how were respondents contacted etc.
  • The background and theoretical grounds of the questionnaire is missing.
  • It is not clear whether the conditions of testing were fulfilled. What is the value of Cronbach Alpha for example for each question? What role it has within the questionnaire? Address also consistency please.
  • The representativeness of data and results is not discussed. Authors should explain whether the sample is representative and to what extent.
  • Is the distribution of gender and other selected characteristics representing the Russian economy? Add that information to Methods please.
  • Table 4 and 5 do not provide information about number of respondents and how was the output created.
  • Sources of other tables are missing as well.
  • Please, clarify importance of outputs and tables provided and add comments regarding implications in each subchapter in results.
  • Discussion does not discuss the results of the paper. Most of it is off topic or just theory. Sometimes not even connected to this study. Author should rewrite this section to match the results and discuss their relevance, validity, applicability, limitations etc.
  • Conclusion is general, only referring to overall situation and theory. It does not conclude the paper with main relevant findings, relevant for readers. Please, address this concern and rephrase your statements. Please, also add to the Conclusions limitations of your paper, future implications, and contribution of your approach to theory and practice. It is still not clear from the current paper.
  • Currently sources are sometimes quite old and new ones needs to be added.
  • The References are not unified and does not fit the standards according to instructions for authors.

Reviewer 4 Report

Dear Authors!

Thank you for taking up a very interesting topic of civil service and young people. It is certainly an issue that deserves investigation.

In my opinion, the paper is characterized by a large number of serious shortcomings. The attempt at multi-faceted analysis of the problem deserves recognition, but it is not a successful attempt.

The list of main shortcomings to be removed is presented below:

  • The aim of the study should be precise and defined in the abstract
  • The abstract should not be written in the first person
  • There are too many keywords, and for example "regional economy" does not appear in the text
  • A definition of the term "young people" is needed. One of the categories existing in the literature should be referred to. Contrary to what the author says "most common self-esteem - people under 30-34" is not enough
  • Contrary to what the author claims, /lines 125-127/ the fact that research was in line with previous examinations does not mean that it is reliable and representative. What about groundbreaking research projects (being not in the line with previous research)?
  • There is methodological chaos. The boundary lines between qualitative and quantitative studies blur
  • From its nature, qualitative research (what the in-depth interviews are) can’t be representative
  • The author uses the terms "interview" and "survey" interchangeably, although these are completely different research techniques, representing different methodological perspectives (quantitative and qualitative). For example, when presenting the results, a survey uses the term interview /lines 220/
  • Hypotheses are not directly derived from the literature
  • The conclusions should be expanded
  • The paper needs proofreading

As a result, I cannot recommend the text for publication.

Best regards,

The reviewer.